# From Clinic to Reality: Integrating Sound Field Testing and Hearing Quality Measures in Cochlear Implant Users

**DOI:** 10.3390/jcm14238430

**Published:** 2025-11-27

**Authors:** Marta Álvarez-Cendrero, Manuel Lazo-Maestre, Serafín Sánchez-Gómez, María A. Callejón-Leblic

**Affiliations:** 1Department of Otolaryngology, Head and Neck Surgery, De La Merced University Hospital, 41640 Osuna, Spain; 2Department of Otolaryngology, Head and Neck Surgery, Virgen Macarena University Hospital, 41009 Seville, Spain; manuel.lazo@faigesco.es (M.L.-M.); mcallejon@us.es (M.A.C.-L.)

**Keywords:** cochlear implant, hearing loss, sound environment audiometric rooms, word recognition score, sound localization, SSQ-12 questionnaire, bimodal fitting

## Abstract

**Background:** Traditional audiological assessment often does not properly reflect the challenges experienced by cochlear implant (CI) users in complex listening environments. Deficits in speech recognition and sound localization ability persist despite clinically satisfactory audiometric thresholds. **Objective:** The objective of this study is to evaluate the impact of acoustic environment complexity on speech recognition and sound localization ability in CI users compared to normal-hearing (NH) individuals and to assess subjective auditory performance using validated questionnaires. **Methods:** Thirty-four unilateral CI users and fifty NH participants were evaluated in a sound environment audiometric room equipped with an eight-speaker 360° array. The variables examined were pure-tone average (PTA), word recognition score (WRS) in quiet and noise (sea, traffic, cafeteria), sound localization ability, and the SSQ-12 auditory quality questionnaire. Correlation, regression, and MANOVA analyses examined associations among objective and subjective outcomes. **Results:** CI users exhibited significantly lower WRS in quiet and noise conditions, reduced sound localization, and decreased SSQ-12 scores compared to NH participants, with performance declining as noise complexity increased. Pre-implant PTA was the strongest predictor of post-implant speech recognition. No significant differences were observed between unilateral CI users and those with bimodal fittings. The SSQ-12 questionnaire correlated with objective measures, supporting its clinical relevance. **Conclusions:** CI users exhibit persistent deficits in ecologically valid listening conditions not demonstrated by standard audiometry. Integrating sound field testing and validated hearing quality questionnaires may provide a more comprehensive assessment to guide personalized rehabilitation and device programming.

## 1. Introduction

Traditional audiological assessment relies on tonal and speech audiometry conducted in soundproof booths [1,2]. These standardized procedures are designed to determine auditory thresholds and quantify speech intelligibility under strictly controlled acoustic conditions, typically in quiet or with stationary background noise [1,3]. While these tests remain essential for diagnostic purposes, growing evidence highlights their inherent limitations in capturing the complexity of real-world listening environments [4,5,6]. This gap is particularly relevant when clinical interest lies in understanding real-life functional hearing.

Consequently, individuals with normal or compensated audiometric thresholds may still experience significant difficulties in complex acoustic environments [7,8], as illustrated by the cocktail party effect [9,10]. This is particularly the case among hearing-impaired (HI) patients adapted with auditory devices, such as hearing aids (HAs) and/or cochlear implants (CIs) [7,11]. Although these devices have experienced substantial technological progress in recent decades, many patients still struggle with speech understanding in dynamic and noisy contexts, such as those with multiple talkers and/or non-stationary noise [9,12,13], (e.g., cafes, traffic, etc.). Additionally, emerging concepts, such as “hidden hearing loss” [14], describe individuals who report speech-in-noise difficulties despite normal standard audiological assessment [15,16,17]. These findings further underline the need for assessments that reflect everyday listening demands.

In light of these limitations, new testing paradigms have emerged to better represent the acoustic complexity of everyday environments [18]. Among these, audiometric booths equipped with multiple loudspeakers distributed across a 360° horizontal plane—referred to as “real sound environment audiometric booths”—allow the recreation of ecologically valid soundscapes. This configuration enables a more comprehensive evaluation of auditory abilities, such as sound localization, perception of competing acoustic signals, and speech discrimination in noise, as well as an objective assessment of hearing assistive devices, such as HAs and CIs [1,2,19].

The American Speech–Language–Hearing Association (ASHA) describes central auditory processing as the set of mechanisms that receive, refine, organize, and interpret information from the peripheral auditory system. These mechanisms support key auditory skills, including auditory discrimination, temporal processing, and binaural integration [20]. In this context, assessments conducted in realistic sound environment booths are particularly valuable, as they allow for the evaluation of crucial auditory abilities—such as sound localization, speech discrimination, and the perception of competing signals—three of the five core behavioral auditory processing skills identified by the ASHA [8,21]. This approach provides a more comprehensive understanding of auditory processing and facilitates the identification of central auditory processing disorders that might otherwise remain undetected [1,22,23].

In addition, self-reported hearing questionnaires serve as valuable complementary tools to assess how individuals perceive their hearing abilities and the effectiveness of hearing devices [24]. These instruments provide multidimensional data that support a more detailed analysis of intervention outcomes [25,26,27,28,29]. Nevertheless, their sensitivity and true correlation with objective audiological measures remain underexplored and require further investigation.

In summary, the limitations of conventional audiological methods in evaluating hearing device performance under realistic listening conditions highlight the need for innovative assessment strategies that more accurately reflect ecologically valid auditory performance [30]. Integrating sound environment audiometric evaluations with validated subjective questionnaires and advanced multidimensional data analysis may offer a more holistic framework to capture both the objective and subjective dimensions of hearing. Within this context, the objectives of the study are (i) evaluating the influence of sound environment complexity on speech recognition and sound localization in CI patients as compared to a control group of NH individuals; (ii) assessing the subjective benefits of hearing assistive devices through validated questionnaires; and (iii) determining their association with objective audiological outcomes.

## 2. Materials and Methods

### 2.1. Design and Participants

From January 2023 to December 2024, data from subjects who were assessed in the audiology unit of a tertiary-care level hospital, the Virgen Macarena University Hospital in Seville (Spain), were included in a case–control study. We studied two patient groups: (i) subjects with bilateral severe or profound sensorineural hearing loss who had received a unilateral cochlear implant (CI) and (ii) a control group of normal-hearing (NH) subjects without any subjective perception of hearing loss. The inclusion criteria for both groups were male and female subjects older than 18. In the NH group, individuals with normal otoscopy, no history of prolonged environmental noise exposure or ototoxic drug use, and pure-tone average (4-PTA: 500, 1000, 2000, and 4000 Hz) below 30 dB HL for individuals under 60 years of age and below 40 dB HL for those aged 60 years or older were considered. In the CI group, unilateral CI users—with or without bimodal adaptation using a hearing aid (HA)—were included only if they achieved ≥50% disyllabic word recognition at 65 dB SPL in free field, tested in the best-aided condition, which represents a clinically accepted indicator of functional benefit. The type of electrode and the electrode insertion technique were documented for all CI users. The sample was additionally categorized according to age using the following ranges: 18–40 years, 41–60 years, and above 60 years. Individuals with cognitive deficits or associated neuropsychiatric pathologies were excluded from this study. The overall study design and participant distribution are illustrated in Figure 1.

This research study was developed following the principles of the Declaration of Helsinki. This study was approved by the Ethics Committee of the Virgen Macarena University Hospital (Seville, Spain) (SICEIA-2024-001950). All patients were informed of the details of the procedures and signed a written informed consent.

### 2.2. Test Material

All tests were conducted in a controlled acoustic environment, specifically in a sound-treated room with internal dimensions of 2.34 × 3.55 × 2.80 m. The experimental procedures were carried out using an AD629 audiometer (Interacoustics A/S, Middelfart, Denmark), calibrated in accordance with IEC 60645; Electroacoustics equipment—Part 6: Instruments for the measurement of otoacoustic emissions. International Electrotechnical Commission (IEC), Geneva, Switzerland, 2022), ISO 389-2; Acoustics—Reference zero for the calibration of audiometric equipment—Part 2: Reference equivalent thresholds sound pressure levels for pure tones and insert earphones. International Organization for Standardization (ISO), Geneva, Switzerland, 1994; and ISO 389-3; Acoustics—Reference zero for the calibration of audiometric equipment—Part 3: Reference equivalent threshold vibratory force levels for pure tones and bone vibrators. International Organization for Standardization (ISO), Geneva, Switzerland, 2016 standards.

Acoustic stimuli were delivered via Genelec 8020D loudspeakers (Iisalmi, Finland). Eight speakers were arranged in a surround configuration around the listener’s position, where the subjects were seated.

The testing room also employed Reaper (Cockos Incorporated, Rosendale, New York, United States) as the digital audio workstation, enabling precise control over devices and sound effects. An external audio interface, the Focusrite Scarlett 18i20 (High Wycombe, England), was used for signal routing across the entire setup.

Additionally, the room was equipped with all necessary materials to perform a wide range of standard audiological assessments, including pure-tone audiometry, speech-in-noise recognition tests, cortical evoked potentials, spatial discrimination tasks, and more.

### 2.3. Data and Outcome Measures

The data collected included age, gender, and audiological outcome measures that evaluated auditory sensitivity, speech intelligibility, sound localization, and subjective quality of hearing. In addition, for subjects with cochlear implants, variables such as the type of electrode used (straight or perimodiolar), the insertion approach (via the round window or cochleostomy), and whether the insertion was complete or incomplete were also analyzed.

Auditory thresholds were assessed from 250 Hz to 8 KHz in a soundproof booth with insert earphones in the NH group and in free field in the CI group. During the experimental procedures, a suprathreshold stimulation level was ensured for each CI participant in order to guarantee activation of the sound processor. Processor activation was verified by monitoring the sound-input indicator lights, when available, depending on the specific brand and model used. All CI users were fitted and clinically programmed by a certified audiologist following standard manufacturer protocols and established fitting guidelines [31,32,33,34,35].

Speech intelligibility in quiet was assessed through the word recognition score (WRS) using a list of 25 disyllabic words (Cárdenas & Marrero, 1994 [36] presented in free field at 65 dBSPL from a loudspeaker at 0° azimuth. Speech intelligibility in noise was assessed by adding three types of competitive noise (sea, traffic, cafeteria) using several simultaneous speakers at 0°, 45°, 90°, 135°, 180°, 225°, 270°, and 315° azimuth with a signal-to-noise ratio (SNR) of +5 dB.

To assess the ability to localize sound, pulsed narrow-band noise (NB) at 250 Hz [37,38] was presented during 200 ms at a fixed level of 65 dB SPL through each speaker in a random order [38,39]. The subjects were positioned at the center of the sound field room and instructed to keep their heads fixed, facing forward. In this position, they were asked to indicate the loudspeaker from which they perceived the sound to originate. A total of 16 stimuli were emitted, two through each of the eight loudspeakers, and the absolute percentage of correct responses was recorded (n° of correct responses × 100/16).

To assess the subjective quality of hearing, all patients were asked to complete the Speech, Spatial, and Qualities of Hearing Scale (SSQ-12) questionnaire [26,40,41]. This questionnaire comprises 12 items, each rated on a scale from 0 to 10 according to the amount of effort required from the subject to complete the task. The final score is determined by calculating the arithmetic mean of the responses. The items are divided into 3 subcategories: the “speech” subdomain evaluates speech ability in background noise and multiple speaker scenarios; the “spatial” subcategory evaluates sound localization ability; and “qualities” addresses a subject’s ability to segregate and identify sounds, as well as the amount of concentration required.

### 2.4. Statistical Analysis

The statistical analysis was conducted using IBM SPSS Statistics version 25 and MatLab 2024b. Descriptive statistics for patient demographics and outcome measures were analyzed for the two groups. The normality of variables was assessed using the Shapiro–Wilk test. Differences in outcome measures, i.e., verbal discrimination in quiet and in competitive noise, sound localization, and the SSQ-12 score between the two groups, were analyzed using the Mann–Whitney U test. The Spearman test was applied to determine possible correlations among these variables and other covariates, such as age and gender. A *p*-value = 0.05 was considered significant in our analysis.

Finally, we constructed a linear regression model by applying stepwise selection methods, initiating the process from an intercept-only model. The response variable was predefined, and the predictor variables were evaluated iteratively through both forward inclusion and backward elimination. Covariate terms were added to (or removed from) the model based on the deviance criterion, which uses a *p*-value of an F statistic to test models with and without a potential term at each step. *p*-values of 0.05 and 0.1 were considered thresholds for adding or removing a term, respectively. Model results were reported using adjusted effect estimates (β), standard errors (SEs), 95% confidence intervals, and *p*-values. The coefficient of multiple determination values (R^2^) was used to assess model fitting.

A MANOVA test was also conducted using a selected subset of independent variables to distinguish between the primary groups: NH, CI users without bimodal adaptation, and CI users with contralateral HA (CI + HA). MANOVA transforms the original data into a new multivariate space defined by canonical variables (CVs)—linear combinations of the independent variables that maximize group separation. This transformed space enables a more efficient classification of observations based on grouping variables. In addition to the graphical clustering results, the outcomes were evaluated using several statistical indicators: Wilks’ Λ, which tests for differences between the group means, χ^2^ (assesses the significance of each canonical subspace), Rao’s F (evaluates whether group means are meaningfully distinct), *p*-values, and effect sizes.

## 3. Results

A total of 84 subjects were included in this study (Table 1). Of these, 50 were normal-hearing (NH) individuals with a mean age of 46.2 ± 15.6 (range: 20–73), of whom 33 (66%) were women, and 34 were cochlear implant (CI) users with a mean age of 56.6 ± 15.9 (range: 26–79), of whom 21 (61.7%) were women. The NH and CI groups were homogeneous in terms of age and sex, i.e., there were no differences in these variables between the two groups (t = 2.947, *p* = 0.004) and (χ^2^ = 0.158, *p* = 0.05), respectively.

In the CI group, half of the users wore the device on the left side, and 15 (44.1%) used a hearing aid in the contralateral ear. There was a full insertion of the electrode in 28 (82.4%) cases, performed through the round window in 23 (67.6%) cases and via cochleostomy in 11 (32.4%) cases. Regarding the type of electrode, 29 (85.3%) were straight, and 5 (14.7%) were perimodiolar electrodes.

### 3.1. Descriptive Analysis

Concerning outcome measures, statistically significant differences were found between NH individuals and (CI) users in pure-tone average (PTA) (t = 11.259, *p* < 0.001) (Figure 2), as well as in WRS in quiet conditions and across all types of competitive noise environments analyzed, with a difference of 15.4% in quiet (t = 10.184, *p* ≤ 0.001), 35.1% for sea noise (U = 94.0, *p* ≤ 0.001), 48.0% for traffic noise (U = 41.0, *p* ≤ 0.001), and 60.7% for cafeteria noise (t = 15.1, *p* < 0.01).

In relation to sound localization ability, a statistically significant difference of 38.4 correct response points was observed between the NH and CI groups (t = 9.1; *p* < 0.01).

Statistically significant differences were found in SSQ-12 scores between the two groups, with a mean score of 7.32 ± 1.18 in the NH compared to 4.46 ± 1.72 in the CI group (t = 44.1; *p* < 0.01).

### 3.2. Correlation Analysis

Table 2 and Table 3 show the Spearman correlation coefficients found between the variables analyzed for the NH and CI groups. Additionally, Appendix A shows the correlation results for the overall sample of subjects.

In the NH group (Table 2), age showed a significant correlation with PTA (r = 0.748; *p* < 0.001) and with WRS in noise, with stronger correlations being found as the complexity of the competitive noise increased for sea (r = −0.189; *p* < 0.001), traffic (r = −0.448; *p* < 0.001), and cafe (r = −0.611; *p* < 0.001). In addition, age was significantly correlated with SSQ-12 scores (r = −0.460; *p* < 0.001). Sound localization showed a significant correlation with WRS under sea competitive noise (r = 0.700; *p* < 0.05). SSQ-12 scores were significantly correlated with PTA (r = −0.334; *p* < 0.05) and with WRS in complex acoustic scenarios such as traffic (r = 0.218; *p* < 0.05) and cafe (r = 0.301; *p* < 0.05).

In the CI group (Table 3), no statistically significant correlations were found between age and any of the other variables. However, PTA was significantly correlated with WRS both in quiet (r = −0.391; *p* < 0.05) and in competitive noise, with stronger associations being found as the acoustic environment became more complex: sea (r = −0.436; *p* < 0.01), traffic (r = −0.545; *p* < 0.001), and cafe (r = −0.573; *p* < 0.001).

When analyzing both groups together (NH and CI) (Appendix A), age was correlated with PTA (r = 0.575; *p* < 0.001), sound localization ability (r = −0.303; *p* < 0.01), and WRS in quiet (−0.218; *p* < 0.05)) and in noise: sea (r= −0.304; *p* < 0.01), traffic (r = −0.424; *p* < 0.001), and cafe (r = −0.491; *p* < 0.001). PTA was also significantly correlated with WRS in quiet (r = −0.678; *p* < 0.001) and noise, i.e., sea (r = −0.737; *p* < 0.001), traffic (=r −0.813; *p* < 0.001), and cafeteria (r = −0.870; *p* < 0.001), and with sound localization (r = −0.638; *p* < 0.001). Furthermore, sound localization ability was, in turn, significantly correlated with WRS in quiet (r = 0.670; *p* < 0.001) and under the three competitive noises: sea (r = 0.493; *p* < 0.001), traffic (r= 0.580; *p* < 0.001), and cafe (r = 0.625; *p* < 0.001).

### 3.3. Regression Analysis

In the NH group (Table 4), speech recognition in quiet was significantly although weakly (R^2^ = 0.224) influenced by age (b = 0.083, CI 95%: [0.035, 0.130], *p* = 0.001) and PTA (b = −0.166, CI 95%: [−0.313, −0.019], *p* = 0.027), while SSQ-12 scores weakly predicted (R^2^ = 0.204) WRS in cafeteria noisy conditions. A stronger effect was found for WRS in the cafeteria, influenced by age (b = −0.358, CI 95%: [−0.692, −0.023], *p* = 0.037) and PTA (b = −1.176, CI 95%: [−2.202, −0.149], *p* = 0.026) with an overall R^2^ equal to 0.411.

In the CI group (Table 5), the WRS outcomes were mainly determined by residual hearing and the pre-implant threshold (b = −0.399, CI 95%: [−0.564, −0.234], *p* ≤ 0.001), with poorer audiometric measures associated with reduced speech recognition in cafeteria scenarios (b = −1.503, CI 95%: [−2.1634, −0.8431], *p* ≤ 0.001). Localization and SSQ-12 scores were not significant predictors in either group.

### 3.4. MANOVA Analysis

As shown in Table 6 and illustrated in Figure 3, the MANOVA model successfully differentiates between the NH and CI groups within this canonical space. The selected independent variables were PTA, WRS_quiet, WRS_cafe, localization, and the total SSQ-12 score. However, upon further analysis, no combination of CVs was able to clearly separate CI users from those using both CI and HA—even when disregarding the distinction from the NH group.

The test revealed a robust and statistically significant effect of group membership on the dependent variables. The first canonical dimension yielded a Wilks’ Λ = 0.153, indicating substantial separation between groups. The associated χ^2^ was 148.13 with 10 degrees of freedom and Rao’s approximate F(2, 81) = 223.6, with *p*-value < 0.001, thus confirming the strength and reliability of the effect. Also, the multivariate effect size (η^2^ = 0.59) suggests that almost 59% of the total variance can be attributed to group differences.

To explore the relationships between the independent variables and the canonical variables, we calculated their correlations. The first CV showed strong correlations with WRS_cafe (r = 0.94, *p* < 0.001) and PTA (r= −0.87, *p* < 0.001) and moderate correlations with the remaining variables (Appendix A).

### 3.5. SSQ-12: Subdomain Analysis

Strong cross-correlations among SSQ-12 subscales (speech, spatial, and quality) were found in the whole sample (Table 7). PTA was negatively correlated with all subdomains of SSQ-12 as well as with the total score. WRS showed significant positive correlations with SSQ-12 subdomains across different listening conditions (quiet, sea, noise, traffic, and cafeteria). Additionally, age was negatively correlated with all SSQ-12 measures.

In the item-level analysis, Question 7 had the greatest influence on auditory abilities, showing the highest predictive value for speech-in-noise discrimination (r = 0.466; *p* < 0.01), whereas Question 6 emerged as the strongest predictor of sound localization ability (r = 0.365; *p* < 0.01).

## 4. Discussion

### 4.1. Differences in Hearing Performance Across Groups

This study reveals pronounced differences between NH and CI users across nearly all audiological and perceptual measures, with CI users demonstrating significantly lower speech recognition, sound localization, and subjective hearing performance (SSQ-12) despite comparable age and sex distributions (Table 1) [42,43]. It is noteworthy that these differences were observed even though the inclusion criteria required CI users to exhibit good performance in conventional audiometric tests (pure-tone audiometry and word recognition in quiet), thus highlighting that impairment may appear in more complex listening situations. The magnitude of these deficits, particularly under conditions of competing noise, corroborates previous evidence emphasizing the enhanced vulnerability of CI users in challenging listening environments [44,45] and underscores the persistent limitations in auditory processing experienced by this population [46,47]. In addition, we did not find significant differences between CI patients and those individuals using bimodal fitting (cochlear implant plus hearing aid) in our cohort (Appendix A). This finding may be related to the high degree of contralateral hearing loss in our patients (Table 1), which could limit the potential benefit of the hearing aid, or the programming asymmetry inherent in fitting two devices that deliver different acoustic and electric signals, thereby complicating the optimization of bimodal stimulation [48]. Although the benefits of binaural hearing are well established in individuals with bilateral cochlear implants, evidence of such benefits in patients with unilateral cochlear implants using a bimodal fitting with a hearing aid remains more limited [49,50].

#### 4.1.1. Speech Recognition

While CI users achieved relatively adequate intelligibility in quiet (83.4% vs. 99.2% in NH), a substantial performance gap emerged as acoustic complexity increased. Performance declined by 35% in sea noise, 48% in traffic, and over 60% in cafeteria noise (Table 1). These findings underscore the limitations of current CI signal processing in segregating speech from background noise and preserving fine temporal cues [47,51,52,53]. Beyond technological constraints, factors such as post-implant neural plasticity [53], duration of device use [54,55], and residual hearing [56,57] likely contribute to these deficits. Moreover, the observed correlation between pre-implant thresholds and noise performance highlights the prognostic significance of residual hearing (Table 5).

#### 4.1.2. Sound Localization

NH participants significantly outperformed CI users, with an average difference of 38.4 points (Table 1). This outcome was expected given the reliance of localization on binaural cues, which are compromised in unilateral CI users or asynchronous bimodal stimulation [51,53]. Although partial benefits have been reported in bimodal listeners, our MANOVA analysis (Appendix A) did not reveal significant differences between unilateral CI and bimodal users, suggesting that the mere addition of a contralateral acoustic device does not guarantee meaningful improvements in spatial hearing performance.

The choice of a 250 Hz NB noise stimulus was made to emphasize temporal and intensity cues while minimizing spectral information. This low-frequency region is classically used to probe ITD sensitivity and to reduce the contribution of high-frequency spectral cues [38,58], providing a controlled way to assess the extent to which CI users can rely on temporal information. However, because many CI recipients—particularly those with shallow insertions or reduced apical stimulation—have limited access to low-frequency cues [37,58], the use of 250 Hz may have accentuated the performance gap between groups.

### 4.2. Subjective Auditory Perception and Correlation with Objective Measures

SSQ-12 scores were markedly higher in NH subjects (Table 1), confirming that objective deficits translate into reduced perceived auditory quality. Strong correlations with intelligibility and localization further support the SSQ-12 as a clinically valuable tool for objective testing [28]. Subscale analyses revealed that the “speech” and “spatial” domains were most strongly associated with noise and localization performance, respectively (Table 7). Item-level findings (Q7: speech in noise, Q6: localization) showed strong predictive value, reinforcing the potential of personalized rehabilitation strategies. Taken together, these results highlight that the SSQ-12 is a guide for target clinical interventions, where item-specific patterns can inform individualized hearing aid fitting, auditory training programs, and counseling approaches aimed at maximizing everyday communicative outcomes [26,59].

### 4.3. Influence of Age

Among NH participants, age correlated with poorer thresholds, reduced speech recognition in noise, and lower SSQ-12 scores (Table 2), consistent with presbycusis-related decline, particularly above 60 years [10,11,60]. In contrast, age was not a significant predictor of auditory outcomes in the CI group, probably due to the fact that implant-related performance overshadowed age-related effects, with residual hearing and electrode insertion having a greater influence on outcomes [61].

### 4.4. Multivariate Analysis and Predictors of Performance

Regression analyses revealed weak contributions of age and auditory thresholds to speech performance in NH listeners (Table 4) [62], while pre-implant thresholds emerged as the strongest predictor of CI outcomes. MANOVA confirmed a clear separation between the NH and CI groups (η^2^ = 0.59) (Table 6), though the model failed to distinguish unilateral from bimodal users, indicating that device combination may not overcome fundamental implant limitations [63]. While whole group-level differences are robust, subject-level improvements derived from bimodal adaptation are more variable [64].

Strong correlations between SSQ-12 scores and localization highlight the value of subjective measures taken together with objective tests [28].

### 4.5. Clinical Implications

The present findings have important implications for clinical practice. First, CI candidates should be counseled about persistent challenges in noise and localization, even with satisfactory performance in quiet [51,61]. Second, continued development of binaural stimulation technologies and noise-reduction algorithms is needed to provide a more natural hearing sensation similar to that of NH [65]. Third, pre-implant residual hearing could guide candidacy and surgical strategies, e.g., full insertion, electrode selection, and electroacoustic stimulation. Finally, subjective measures, such as the SSQ-12, should be integrated into evaluations in both NH and CI groups to capture daily life difficulties beyond laboratory tests [25,66]. The clinical application of more ecologically speech-in-noise and sound localization tests emulating complex listening environments, such as those of real life, could guide more optimized and personalized therapeutic strategies and fitting procedures [64,67].

### 4.6. Limitations and Implications for Future Research

Despite certain limitations, this study has the advantage of including a relatively large, well-characterized patient cohort, a factor that considerably strengthens the robustness, statistical power, and external validity of our findings compared to most previous studies [13,32,68]. The inclusion of a broad range of adult CI users with different electrode types and fitting configurations provides a more representative overview of real-world auditory outcomes, in line with recent large-scale reviews on post-implant variability [8,24].

However, several limitations should be acknowledged. Although the cohort was larger than in many comparable studies [1,9,69], generalizability remains limited to populations with similar demographic and clinical characteristics.

Furthermore, the cross-sectional design does not permit longitudinal assessment of auditory adaptation or neural plasticity processes after implantation, aspects that have been recently highlighted as crucial determinants of auditory rehabilitation success [18,70].

Additionally, factors such as duration of CI use, speech processor mapping strategies, and post-operative rehabilitation intensity were not examined, despite their documented influence on long-term auditory and speech-in-noise outcomes [31,32]. Moreover, an in-depth assessment of MCL/T settings was not performed, and programming was not adjusted based on the study results. Instead, fitting followed standardized clinical protocols guided by audiogram and speech intelligibility outcomes; thus, the reported results reflect performance under routine clinical practice. These variables were analyzed only descriptively and not used to optimize fitting, which constitutes an additional limitation. Developing advanced fitting protocols incorporating spatial localization and speech-in-noise measures remains a necessary next step, as our findings suggest significant potential for individualized optimization even in cohorts with substantial benefits (≥50% intelligibility in quiet).

Future research should focus on longitudinal analyses to capture post-implant performance trajectories and adaptive plasticity mechanisms, as recommended in recent meta-analyses [8,13]. Comparative investigations of coordinated binaural processing versus independent device use are warranted to optimize bilateral and bimodal stimulation strategies. Future studies should therefore incorporate a broader range of stimulus frequencies to better understand how frequency content interacts with device configuration and residual acoustic hearing in shaping localization abilities. Furthermore, the development of evidence-based auditory rehabilitation programs targeting speech-in-noise discrimination, spatial orientation, and cognitive listening effort—potentially enhanced through machine learning-driven fitting algorithms and AI-assisted implant processors—represents a promising avenue for future innovation [24,33].

## 5. Conclusions

This study demonstrates that even when cochlear implant (CI) users achieve satisfactory audiometric thresholds and word recognition scores in quiet conditions, they continue to exhibit pronounced deficits in speech discrimination within complex acoustic environments and in sound localization when compared to normal-hearing (NH) individuals. These objective impairments are also consistently reflected in lower self-reported auditory quality, as evidenced by significantly reduced SSQ-12 scores.

Multivariate analyses further confirmed that conventional audiological assessments fail to capture the functional challenges experienced by CI users in realistic, dynamic listening conditions. Among the analyzed variables, pre-implant audiometric thresholds emerged as the strongest predictor of speech performance in noise, reinforcing previous evidence on the prognostic value of residual hearing for post-implant outcomes.

Additionally, no significant differences were observed between unilateral CI users and those with bimodal fittings, suggesting that the addition of a contralateral hearing aid alone does not consistently yield functional improvements under ecologically demanding conditions—particularly when the non-implanted ear presents severe-to-profound hearing loss.

Taken together, these findings align with recent meta-analytic evidence highlighting the persistent gap between laboratory-based and real-world auditory performance in IC users [8,13]. They emphasize the importance of integrating ecologically valid sound-field testing and patient-reported outcome measures (such as the SSQ-12) into routine clinical evaluation. Moreover, the results underscore the broader need for surgical and programming strategies that preserve residual hearing, optimize binaural stimulation, and guide the development of advanced signal-processing algorithms and personalized auditory rehabilitation programs to improve speech-in-noise perception and spatial hearing abilities.

## Figures and Tables

**Figure 1 jcm-14-08430-f001:**
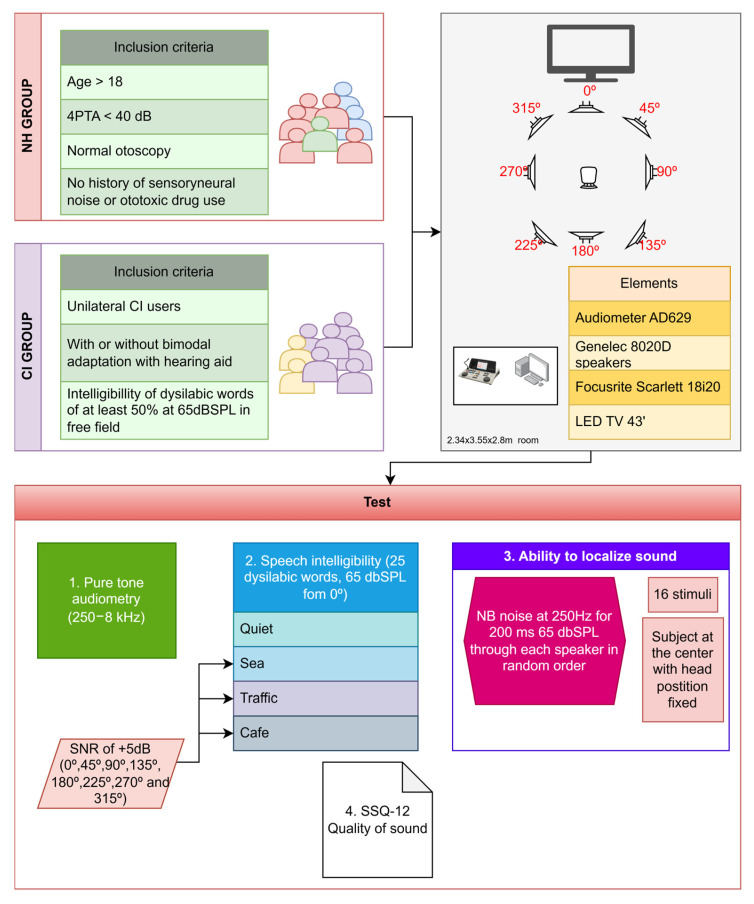
Experimental workflow: tests and procedures for both groups.

**Figure 2 jcm-14-08430-f002:**
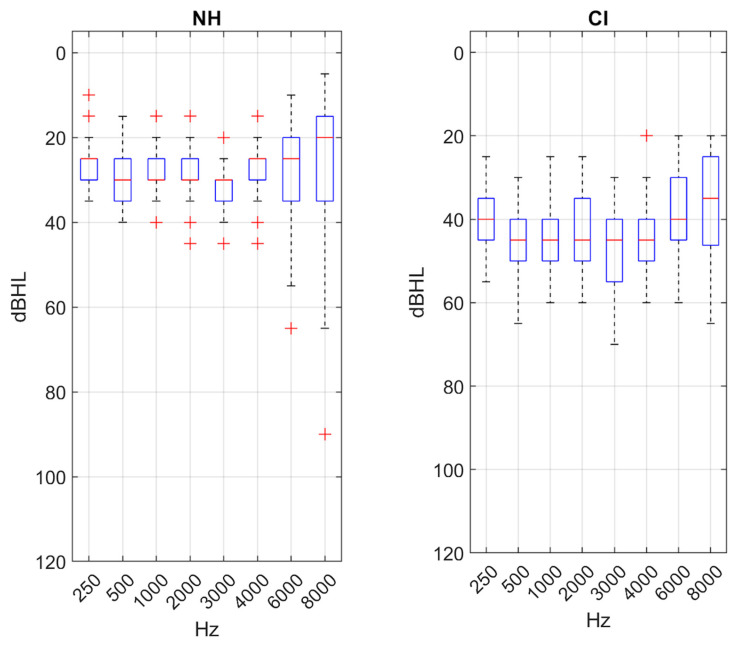
Boxplots representing values of the PTA for each main group.

**Figure 3 jcm-14-08430-f003:**
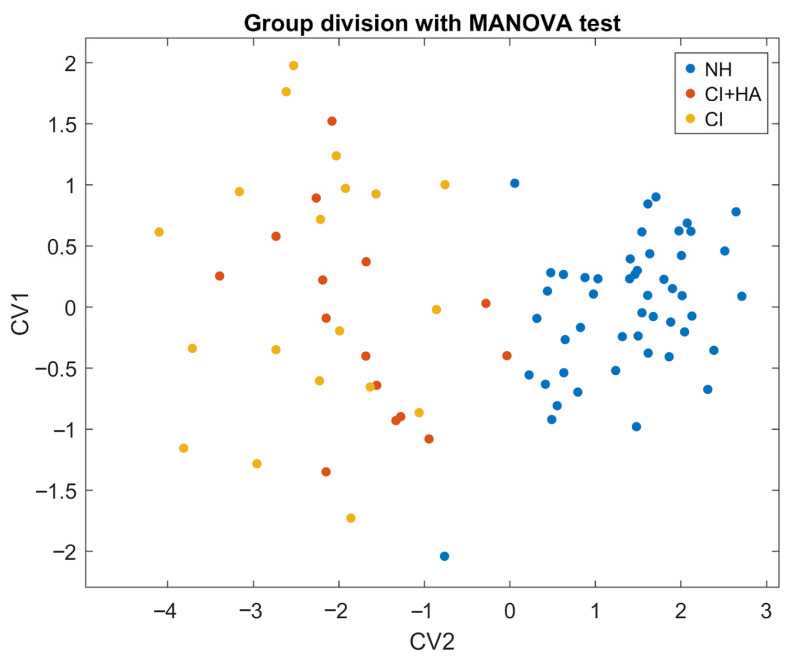
Scatter plot of MANOVA results. Canonical variables combine predictors to maximize group separation. CV1 provides the strongest NH-CI discrimination, with CV2 adding orthogonal information. The plot shows this 2D canonical space, with colored vectors indicating each predictor’s correlations with CV1 and CV2.

**Table 1 jcm-14-08430-t001:** Distribution of demographic characteristics and audiological outcome measures. Numerical variables are displayed as mean +/− SD when normally distributed; otherwise, they are displayed as median (IQR). Qualitative variables are expressed as percentages.

	NH (n = 50)	CI (n = 34)	*p*-Value
Gender			0.69
Women	33 (66.0%)	21 (61.8%)	
Men	17 (34.0%)	13 (38.2%)	
$\bar{x}$ age ± SE	46.2 ± 15.6 [20–73]	56.6 ± 15.9 [26–79]	0.348
Implanted ear	-		-
Right		17 (50%)	
Left		17 (50%)	
Full electrode insertion	-	27 (79.4%)	-
Type of insertion	-		-
Round window		22 (64.7%)	
Cochleostomy		11 (32.4%)	
Type of implant			-
Straight		30 (88.2%)	
Perimodiolar		4 (11.8%)	
Bimodal fitting		15 (44.1%)	-
PTA (dBHL)	28.5 ± 5.1 [15–40]	44.3 ± 7.7 [25–61.3]	<0.01
Pre-CI PTA (dBHL)			
Ipsilateral		91.5 ± 22.7 [53–120]	
Contralateral		96.0 ± 22.8 [72–120]	
WRS (%) in quiet	99.2 ± 2.0 [90–100]	83.4 ± 10.7 [65–100]	<0.01
WRS (%) in noise (sea)	88.3 ± 8.5 [68–100]	53.2 ± 21.4 [0–90]	<0.01 ^ε^
WRS (%) in noise (traffic)	84.1 ± 11.3 [52–100]	36.1 ± 23.2 [0–80]	<0.01 ^ε^
WRS (%) in noise (cafeteria)	73.9 ± 16.6 [48–100]	16.2 ± 18.1 [0–70]	<0.01
Sound localization (%)	56.9 ± 22.7 [6.3–100]	18.7 ± 11.8 [0–43.8]	<0.01
SSQ-12	7.3 ± 1.2 [2.3–9.1]	4.5 ± 1.8 [1.1–7.6]	<0.01

$\bar{x}$ = arithmetic average, SE = standard deviation. ^ε^
*p*-value as determined by the Mann–Whitney U test; Me = median, IQR = interquartile range.

**Table 2 jcm-14-08430-t002:** Spearman correlation coefficients for the NH group. * *p*-value < 0.05; ** *p* < 0.01; *** *p* < 0.001.

	Age	PTA	WRS Quiet	WRS Sea	WRS Traffic	WRS Cafe	Sound Localization	SSQ-12
Age	1	0.748 ***	0.309 *	−0.189 ***	−0.448 ***	−0.611 ***	−0.039	−0.460 ***
PTA	0.748 ***	1	0.108	−0.321 *	−0.475 ***	−0.667 ***	−0.213	−0.334 **
WRS quiet	0.309 *	0.108	1	0.038	−0.038	−0.141	0.201	−0.106
WRS sea	−0.189	−0.321 *	0.038	1	0.700 ***	0.485 ***	−0.270 *	0.007
WRS traffic	−0.448 ***	−0.475 ***	−0.038	0.700 ***	1	0.704 ***	−0.173	0.281 *
WRS cafe	−0.611 ***	−0.667 ***	−0.141	0.485 ***	0.704 ***	1	0.011	0.301 *
Sound localization	−0.039	−0.213	0.201	−0.270*	−0.173	0.011	1	0.061
SSQ-12	−0.460 ***	−0.334 *	−0.106	0.007	0.218 *	0.301 *	0.061	1

**Table 3 jcm-14-08430-t003:** Spearman correlation coefficients for the CI group. * *p*-value < 0.05; ** *p* < 0.01; *** *p* < 0.001.

	Age	PTA	WRS Quiet	WRS Sea	WRS Traffic	WRS Cafe	Sound Localization	SSQ-12
Age	1	0.224	−0.182	−0.056	−0.140	−0.219	−0.252	−0.206
PTA	0.224	1	−0.391 *	−0.436 **	−0.545 ***	−0.573 ***	−0.175	−0.135
WRS quiet	−0.182	−0.391 *	1	0.463 **	0.322	0.129	0.047	−0.184
WRS sea	−0.056	−0.436 **	0.463 **	1	0.795 ***	0.577 ***	0.387 *	−0.166
WRS traffic	−0.140	−0.545 ***	0.322	0.795 ***	1	0.801 ***	0.261	−0.127
WRS cafe	−0.219	−0.573 ***	0.129	0.577 ***	0.801 ***	1	0.238	−0.023
Sound localization	−0.252	−0.175	0.047	0.387 *	0.261	0.238	1	0.017
SSQ-12	−0.206	−0.135	−0.184	−0.166	−0.127	−0.023	0.017	1

**Table 4 jcm-14-08430-t004:** Multiple regression analysis for the NH group. Independent variables: age, sex, PTA, WRS_quiet, WRS_sea, WRS_traffic, WRS_cafe, and localization ability.

	Adjusted b (mean ± SE)	95% CI	*p*-Value	R^2^
SSQ-12				0.224
WRS (noise—cafe)	0.040 ± 0.011	[0.019, 0.064]	<0.001	
WRS (quiet)				0.204
Age	0.083 ± 0.024	[0.035, 0.130]	0.001	
PTA	−0.166 ± 0.073	[−0.313, −0.019]	0.027	
WRS (noise—cafe)				0.411
Age	−0.358 ± 0.167	[−0.692, −0.023]	0.037	
PTA	−1.176 ± 0.510	[−2.202, −0.149]	0.026	
Localization	-	-	-	0

**Table 5 jcm-14-08430-t005:** Multiple regression analysis for the CI group. Independent variables: age, sex, PTA, WRS_quiet, WRS_sea, WRS_traffic, WRS_cafe, localization, implanted ear, full insertion, insertion type, bimodal fitting, months post-implantation, PTA_pre-CI_Ipsilateral, and PTA_pre-CI_Contralateral.

	Adjusted b (mean ± SE)	95% CI	*p*-Value	R^2^
SSQ-12	-	-	-	0
WRS quiet				0.5102
PTA4_pre_CONTRA	−0.399 ± 0.0799	[−0.564, −0.234]	<0.001	
WRS cafe				0.402
PTA	−1.503 ± 0.3241	[−2.1634, −0.8431]	<0.001	
Localization	-	-	-	0

**Table 6 jcm-14-08430-t006:** MANOVA test results in the main canonical dimensions. The multivariate effect size is 0.5881.

	Wilks’ Lambda	χ^2^ (dof)	Rao’s F (dof)	*p*-Value
Canonical Dimension 1	0.153	148 (10)	223 (2.81)	>0.001
Canonical Dimension 2	0.91631	7 (4)	3.7 (2.81)	0.14

**Table 7 jcm-14-08430-t007:** Spearman correlation coefficients for the overall group. * *p*-value < 0.05; ** *p* < 0.01; *** *p* < 0.001.

	Speech SSQ 12	Spatial SSQ 12	Qualities SSQ 12	SSQ-12_Total
PTA	−0.755 ***	0.852 ***	−0.477 ***	−0.638 ***
Localization	0.576 ***	0.526 ***	0.297 *	0.536 ***
WRS Quiet	0.548 ***	0.539 ***	0.319 *	0.514 ***
WRS Sea	0.556 ***	0.518 ***	0.345 **	0.504 ***
WRS Traffic	0.711 ***	0.707 ***	0.457 ***	0.642 ***
WRS Cafeteria	0.721 ***	0.713 ***	0.504 ***	0.664 ***
Age	−0.508 ***	−0.572 ***	−0.363 **	−0.448 ***
Speech SSQ 12	1	0.852 ***	0.759 ***	0.965 ***
Spatial SSQ 12	0.852 ***	1	0.657 ***	0.905 ***
Qualities SSQ 12	0.759 ***	0.657 ***	1	0.853 ***
SSQ-12_Total	0.965 ***	0.905 ***	0.853 ***	1

## Data Availability

The original contributions presented in the study are included in the article (and Appendix A), further inquiries can be directed to the corresponding authors.

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
