# Peer review of "From Clinic to Reality: Integrating Sound Field Testing and Hearing Quality Measures in Cochlear Implant Users"

_jcm, 2025, doi:10.3390/jcm14238430_

Round 1

Reviewer 1 Report

Comments and Suggestions for Authors

Introduction
The introduction is of sufficient length, but it could be restructured
and refined for clarity. The authors should consider citing more recent
and relevant publications, especially prior systematic work on the field.

Materials and Methods
The methodology appears poor. In my opinion tthere is a lack of substantial information. There is no refer on the activation and the control of the processor. O.k, the authors provide important information about the effects of the implantation, but how can we be sure, if there is no classification of the patients based on the audiological benefit of them? Any reference to MC- or T-value is also helpful.   The section would benefit from stronger bibliographic
support. 

Results
The results are presented clearly, but their impact could be enhanced by
including additional tables or figures. This would improve the clarity
of the data presentation and allow readers to evaluate the findings more
effectively.

Discussion
The discussion is well written overall. 
The authors could also emphasize the significance of their large patient cohort
and discuss possible limitations of the study, as well as directions for
future research.

Conclusions
The conclusions are consistent with the evidence and arguments
presented, and they address the main research question. Nonetheless,
they would be strengthened by explicitly situating the findings within
the broader landscape.

Reviewer 2 Report

Comments and Suggestions for Authors

The authors compared 34 unilateral CI patients and 50 normal hearing controls using more demanding audiology tests (noise, location etc) than eg speech audiometry in quiet. As well known, unilateral CI patients (with or without hearing aid on the other ear) were found to perform worse in the employed tests. It would have been interesting to know the actual hearing level of the non implanted ear, as the hearing aid in the opposite ear did not add any hearing gain.

The authors must explain how 40db normal subjects are included as that group was expected to be better than 30db.

The first lines in the results section is an instruction from the journal and should be omitted.

Reviewer 3 Report

Comments and Suggestions for Authors

This is a very important work in clinical work. The introduction is too long and you can omit many details and focus on the final aim of the work.

You introduced the type of electrode and type of surgery as confounding factors. In many articles you can find that there are no clear , or at best marginal differences, on the long term follow up of patients. I think you can just omit this in your analysis.

You tested NH persons with an insert earphone and the CI subjects in a free field. Did you mask the other normal side, if not the CI patients are at significant disadvantage in sound localization tasks and this is well known in single ear testing. 

You tested for sound localization with a 250 Hz  narrow band noise, this is quite low especially that many CI have some problems with low frequencies especially with short electrodes or partial insertions. If you have used a higher frequency you might have different results ? Please comment

You mentioned that pre-implant thresholds are a positive predictor for a better performance. Did any of your patient undergo a hearing preservation implantation with EAS ?

One important factor that must be highlighted is the quality and duration of rehabilitation [which are typically lacking in many adult CI patients]. This can profoundly affect their outcomes.

Your conclusion is very valid and significant. The only point you might need to elaborated (if possible) is the lack of advantage of bimodal stimulation in your CI users especially as I couldn't find data on their contralateral thresholds [aided and unaided]

Round 2

Reviewer 1 Report

Comments and Suggestions for Authors

The text has been substantially improved. In this case, I do not have any specific recommedations. 

Comments on the Quality of English Language

This version can be published.  The quality of the language is o.k!

Reviewer 3 Report

Comments and Suggestions for Authors

Thank you for addressing the raised issues